# Effects of the Feeding Rate on Growth Performance, Body Composition, and Hematological Properties of Juvenile Mandarin Fish *Siniperca scherzeri* in a Recirculating Aquaculture System

**Yi-Oh Kim [1], Sung-Yong Oh [2],\* and Taewon Kim [3],[4],\***

[1]  Chungcheongbuk-do Inland Fisheries Research Institute, Chungju 27432, Korea; kimio@korea.kr
[2]  Marine Bio-Resources Research Unit, Korea Institute of Ocean Science & Technology, Busan 49111, Korea
[3]  Program in Biomedical Science and Engineering, Inha University, Inha-ro 100, Michuhol-gu, Incheon 22212, Korea
[4]  Department of Ocean Sciences, Inha University, Inha-ro 100, Michuhol-gu, Incheon 22212, Korea
\*  Correspondence: syoh@kiost.ac.kr (S.-Y.O.); ktwon@inha.ac.kr (T.K.)

**Abstract:** The effects of feeding rate (0.5%, 1.0%, 1.5%, 2.0%, 2.5%, and 3.0% body weight [BW] per day [BW day$^{-1}$]) on the growth, body composition, and blood properties of juvenile mandarin fish, *Siniperca scherzeri*, (initial BW 18.4 ± 0.2 g) were investigated in a water recirculating aquaculture system. Triplicate groups of fish were fed an experimental diet (55.4% protein, 14.1% lipid) for 10 weeks. Weight gain and specific growth rate increased with increasing feeding rates of up to 2.5% BW day$^{-1}$, after which no significant increase in growth was observed. Feed efficiency, protein efficiency ratio, and protein retention were not significantly different among the fish fed 1–2.5% BW day$^{-1}$ but decreased significantly in those fed 3.0% BW day$^{-1}$. The lipid content of fish fed 2.5% BW day$^{-1}$ was significantly higher than that at 0.5–1.5% BW day$^{-1}$. The total plasma cholesterol content was significantly lower in fish fed 0.5 BW day$^{-1}$ than fish provided with other feeding rates. Based on the growth, feed efficiency, body composition, and blood content analyses, including regression analysis, the optimal feeding rate for juvenile *S. scherzeri* weighing between 18 g and 54 g was estimated at 1.88–2.80% BW day$^{-1}$ depending on weight gain, specific growth rate, and feed efficiency under 26.9 °C conditions.

**Keywords:** mandarin fish; *Siniperca scherzeri*; feeding rate; growth; body composition; hematology

## 1. Introduction

In Korea, China, and Vietnam, fish of the genus *Siniperca* [1,2] are commercially valuable because the meat is palatable [1–4]. However, they are challenging to farm with formulated fish feed due to their strong preference for live prey [5,6], and thus the commercial supply of *S. scherzeri* is limited [4,7]. Efforts to develop feed formulations for juvenile *S. scherzeri* by applying the latest feed technologies are continuing [3,8–10].

Although the growth of farmed fish is affected by several factors, an optimized feeding regimen is considered to be crucial for the maximum growth of the species [11,12]. In such regimes, maximum fish growth and feed efficiency can be achieved by providing the optimal amount of feed based on the growth stage of the fish and the farming conditions [13,14]. If the feeding regimen is not optimized for the farmed species, feeding can be either insufficient or excessive. Overfeeding leads to food wastage and inefficient nutrition, resulting in financial losses, while insufficient feeding may lead to undergrowth due to a failure to satisfy the nutritional requirements for maximum fish growth [15–17]. Therefore, the optimal feeding rate for maximum growth and feed efficiency that increases the productivity of the aquaculture must be identified [14,18,19]. The optimal feeding rate varies with the fish species, size, feed composition, and culture conditions [20], which explains the need

to investigate the effects of different feeding rates on fish productivity and disseminate the findings. Some studies on the feeding period of *S. scherzeri* have been conducted to determine the optimal feeding regimen [4], whereas no studies have been conducted on the effects of changing the feeding rate. In this study, we investigated the effects of the feeding rate on the growth, body composition, and hematological properties of juvenile *S. scherzeri* to provide information for those developing practical feeding regimens for *S. scherzeri* farming.

## 2. Materials and Methods

### 2.1. Fish and Rearing Conditions

Juvenile *S. scherzeri* were reared for experiments at the Chungcheongbuk-do Inland Fisheries Research Institute, Chungju, Korea. The experiments were performed in a small recirculation culture system comprising of a submerged nitrification filter (3000 L, specific surface area of filter media 200 $m^2$ $m^{-3}$), a foam separator (100 L, retention time 90 s), and 18 circular fiberglass culture tanks (300 L, flow rate 30 L $h^{-1}$). Twenty *S. scherzeri* (average weight of $18.4 \pm 0.2$ g) were randomly housed in each circular culture tank and acclimated for 2 weeks prior to the experiment. During the acclimation period, the fish were fed to visual apparent satiation with the experimental feed twice daily (09:00 and 17:00). Each culture tank was aerated to provide sufficient dissolved oxygen. During the experimental period, the dissolved oxygen, temperature, ammonia, nitrite, and pH levels were monitored daily, and the water temperature was maintained at $26.9 \pm 0.7$ °C. Dissolved oxygen was kept at over 7.1 mg $L^{-1}$, the ammonia was less than 0.75 mg $L^{-1}$, the nitrite was less than 0.18 mg $L^{-1}$, and the pH was in the range 6.7–7.4.

### 2.2. Experimental Design

Mandarin fish groups were allocated at one of the following six feeding rates: 0.5%, 1.0%, 1.5%, 2.0%, 2.5%, and 3.0% BW $day^{-1}$ for 10 weeks. Each feeding rate was randomly assigned to triplicate tanks, each containing 20 juveniles. The total feed amount for each day was divided into two equal portions, which were given to the fish at 09:00 and 17:00 each day. The actual amount of daily food provided was calibrated using the dry weight of leftover food collected from the floor of the tanks 30 min after feeding. The amount of feed was carefully calibrated to minimize leftover feed on the tank floor.

### 2.3. Preparation of Experimental Diets

Table 1 shows the ingredients and chemical compositions of the experimental feed. The anchovy fish meal was the primary protein source, and equal proportions of squid liver oil and soybean oil were added as the lipid source [9]. To prepare the experimental diet, all the dry ingredients were finely ground in a laboratory mill (HKPM-100, HanKook Power System, Seoul, Korea) and well-mixed by a mixer (NVM-18, Daeyung Bakery Machinery Co. Ltd., Seoul, Korea). Squid liver oil, soybean oil, and double-distilled water were then added into the mixture and blended. The well-mixed final moist mash was made into pellets of 3 mm in diameter by a meat chopper machine (SMC-32, SL Co., Incheon, Korea). The produced pellets were dried at room temperature overnight and stored at −30 °C until use.

**Table 1.** Ingredients and proximate composition (%) of the experimental diet.

| Ingredients | Composition (%) |
|---|---|
| Anchovy fish meal [a] | 76.0 |
| Corn gluten meal [b] | 2.8 |
| Potato-starch | 10.5 |
| Squid liver oil + soybean oil | 8.1 |
| Vitamin premix [c] | 1.0 |
| Mineral premix [d] | 1.0 |
| Vitamin C | 0.3 |
| Vitamin E | 0.2 |
| Choline salt | 0.1 |
| Nutrient contents (dry matter basis) | |
| Crude protein (%) | 55.4 |
| Crude lipid (%) | 14.1 |
| Ash (%) | 11.5 |

[a] Pesquera Bahia Caldera, Caldera, Chile. Fishmeal composition (% dry matter): crude protein, 67.3; crude lipid, 8.6. [b] WooSung Feed Corporation, Daejeon, South Korea. Corn gluten meal composition (% dry matter): crude protein, 66.1; crude lipid 2.8. [c] Vitamin premix contained the following ingredients (g kg$^{-1}$ premix), which were diluted in cellulose: thiamin hydrochloride, 2.7; riboflavin, 9.1; pyridoxine hydrochloride, 1.8; niacin, 36.4; Ca-D-pantothenate, 12.7; myo-inositol, 181.8; D-biotin, 0.27; folic acid, 0.68; p-aminobenzoic acid, 18.2; menadione, 1.8; retinyl acetate, 0.73; cholecalciferol, 0.003; cyanocobalamin, 0.003. [d] Mineral premix contained the following ingredients (g kg$^{-1}$ premix): $MgSO_4 \cdot 7H_2O$, 80.0; $NaH_2PO_4 \cdot 2H_2O$, 370.0; KCl, 130.0; Ferric citrate, 40.0; $ZnSO_4 \cdot 7H_2O$, 20.0; Ca-lactate, 356.5; CuCl, 0.2; $AlCl3 \cdot 6H_2O$, 0.15; KI, 0.15; $Na_2Se_2O_3$, 0.01; $MnSO_4 \cdot H_2O$, 2.0; $CoCl_2 \cdot 6H_2O$, 1.0.

### 2.4. Fish Measurement and Body Condition

The length and mass of each fish was measured at the beginning and end of the experiment; all fish were starved for 24 h prior to measuring to prevent variations due to the weight of the stomach contents. The fish were anesthetized with 2-phenoxyethanol solution (150 mg L$^{-1}$, Sigma, St. Louis, MO, USA) and measured for standard length ($\pm 0.01$ cm) and wet mass ($\pm 0.01$ g). At the end of the experiment, 9 fish (3 fish per tank) in each treatment group were randomly sacrificed by anesthetization in 150 mg L$^{-1}$ 2-phenoxyethanol solution for 1 min and stored in a $-40$ °C freezer until the analysis of body composition. The body composition was analyzed as per standard procedures [21]: the crude protein content of the fish was analyzed using the Auto Kjeldahl System (Buchi B-324/435/412, Switzerland; Metrohm 8-719/806, Switzerland), and the water content was measured by weighing the whole body before and after drying in an oven at 105 °C for 24 h. Crude fat was measured according to the ether-extraction method, and crude ash was measured after burning the fish at 600 °C for 4 h.

### 2.5. Analysis of Blood Component and Morphological Indices

To investigate the influence of the feeding rate on blood components, 15 fish (5 fish per tank) in each treatment group were randomly sacrificed at the end of the experiment by the method identical to the body composition sampling. Before the fish were sacrificed, the blood samples of the fish were individually collected from the tail arteries using a heparinized syringe. Hematocrit (HCT) was measured from whole blood, and at least 0.5 mL of serum was collected after centrifugation at $8870 \times g$ for 5 min. Using DRI-CHEM NX500i (Fujifilm Co., Tokyo, Japan), serum total protein (TP), total cholesterol (TCHO), glutamic oxaloacetic transaminase (GOT), glutamic pyruvic transaminase (GPT), high-density lipoprotein cholesterol (HDLC), and glucose (GLU) were measured. Immediately after blood collection, the sacrificed fish were also dissected to determine the weights of the liver and viscera to measure the hepatosomatic index (HSI) and viscerosomatic index (VSI) as morphological indices.

### 2.6. Statistical Analysis

All statistical analyses, including one-way ANOVA, were performed using SPSS Ver. 20 (SPSS Inc., Chicago, IL, USA). Tukey's multiple range test was applied to verify the

significance of the means at the 95% confidence level. The estimation of the optimum feeding rate was determined by the application of quadratic broken-line models for weight gain (WG) and specific growth rate (SGR) using a nonlinear least-squares function, and a quadratic model (second-order polynomial) for feed efficiency (FE) using a linear model function with the standard library of R 3.0.1 [22] from the method used by Lee et al. [23].

## 3. Results

WG ($F = 118.808$, $df = 5$, $p < 0.0001$) and SGR ($F = 213.376$, $df = 5$, $p < 0.0001$) were significantly different among the groups with different feeding rates. Specifically, the fish in the groups allocated the 2.5% and 3.0% feeding rates had significantly higher WG and SGR than the other groups (i.e., 0.5%, 1.0%, 1.5%, and 2.0%), and the differences between the 2.5% and 3.0% feeding groups were not significant (Table 2). Feed consumption ($F = 1010.287$, $df = 5$, $p < 0.0001$) significantly increased as the feeding rate increased (Table 3). FE ($F = 15.354$, $df = 5$, $p < 0.0001$), and the protein efficiency ratio (PER) ($F = 14.883$, $df = 5$, $p < 0.0001$) was significantly different among the different feeding rate groups. Although FE and PER were not significantly different between The 1.0%, 1.5%, 2.0%, and 2.5% groups, the 1.5% group showed significantly higher FE and PER than both the 0.5% and 3.0% groups (Table 3). The protein retention (PR) ($F = 14.372$, $df = 5$, $p < 0.0001$) of the 1.0–3.0% groups was significantly higher than that of the 0.5% group (Table 3).

**Table 2.** Survival, weight gain, and specific growth rate of juvenile mandarin fish *Siniperca scherzeri* fed the experimental diets with various feeding rates for 10 weeks *.

| Feeding Rate (% BW$^{-1}$) | Initial Mean Weight (g) | Final Mean Weight (g) | Survival (%) | WG (%) [1] | SGR (% Day$^{-1}$) [2] |
|---|---|---|---|---|---|
| 0.5 | 18.4 ± 0.12 | 20.9 ± 0.63 [a] | 100 ± 0.0 | 9.7 ± 2.86 [a] | 0.13 ± 0.04 [a] |
| 1.0 | 18.2 ± 0.14 | 31.5 ± 0.40 [b] | 100 ± 0.0 | 64.5 ± 1.01 [b] | 0.71 ± 0.01 [b] |
| 1.5 | 18.2 ± 0.09 | 45.0 ± 0.92 [c] | 100 ± 0.0 | 135.4 ± 4.06 [c] | 1.22 ± 0.02 [c] |
| 2.0 | 18.6 ± 0.08 | 48.2 ± 1.15 [c] | 100 ± 0.0 | 147.3 ± 6.21 [c,d] | 1.29 ± 0.04 [c,d] |
| 2.5 | 18.4 ± 0.35 | 53.9 ± 1.00 [d] | 100 ± 0.0 | 179.9 ± 10.76 [e] | 1.47 ± 0.05 [e] |
| 3.0 | 18.7 ± 0.04 | 54.0 ± 1.29 [d] | 100 ± 0.0 | 175.5 ± 7.17 [d,e] | 1.45 ± 0.04 [d,e] |

* Values (mean ± SE of three replicate groups) with different superscripts in the same column are significantly different ($p < 0.05$). Weight gain (WG) and specific growth rate (SGR). [1] Weight gain (%) = 100 × (final body weight − initial body weight)/initial body weight. [2] Specific growth rate = 100 × (Ln final weight of fish − Ln initial weight of fish)/days of feeding trial.

**Table 3.** Feed consumption, feed efficiency, protein efficiency ratio, and protein retention of juvenile mandarin fish *Siniperca scherzeri* fed the experimental diets with various feeding rates for 10 weeks *.

| Feeding Rate (% BW$^{-1}$) | Feed Consumption (g Fish$^{-1}$) | FE (%) [1] | PER [2] | PR [3] |
|---|---|---|---|---|
| 0.5 | 4.8 ± 0.07 [a] | 38.6 ± 10.74 [a] | 0.69 ± 0.19 [a] | 10.2 ± 4.43 [a] |
| 1.0 | 16.0 ± 0.05 [b] | 77.2 ± 1.59 [b,c] | 1.38 ± 0.03 [b,c] | 27.0 ± 0.42 [b] |
| 1.5 | 28.7 ± 0.27 [c] | 89.9 ± 2.15 [c] | 1.61 ± 0.04 [c] | 30.1 ± 1.40 [b] |
| 2.0 | 33.9 ± 0.60 [d] | 84.7 ± 1.97 [b,c] | 1.52 ± 0.04 [b,c] | 29.7 ± 0.72 [b] |
| 2.5 | 45.4 ± 0.62 [e] | 76.3 ± 1.96 [b,c] | 1.37 ± 0.04 [b,c] | 27.5 ± 0.51 [b] |
| 3.0 | 55.2 ± 1.11 [f] | 62.4 ± 2.44 [b] | 1.12 ± 0.04 [b] | 22.8 ± 0.99 [b] |

* Values (mean ± SE of three replicate groups) with different superscripts in the same column are significantly different ($p < 0.05$). Feed efficiency (FE), protein efficiency ratio (PER) and protein retention (PR). [1] Feed efficiency (%) = 100 × fish wet weight gain/feed intake (dry matter). [2] PER = weight gain of fish/protein consumed. [3] PR = protein gain of fish/protein consumed.

The condition factor (CF) differed significantly among the groups ($F = 54.776$, $df = 5$, $p < 0.0001$); especially that of the 3.0% group, which was significantly higher than in the other groups, and the CF increased as the feeding rate increased (Table 4). The HSI was also significantly different among the feeding groups ($F = 6.956$, $df = 5$, $p = 0.003$); in the 1.5–3.0% groups, especially, the HSI was significantly higher than that in the 0.5% group, whereas the 1.0–3.0% groups were not significantly different with regards to HSI (Table 4).

The VSI ($F$ = 7.681, *df* = 5, *p* = 0.0002) also differed significantly among the groups. While the VSI of the 2.0–3.0% groups was significantly higher than that of the 0.5% group, there was no significant difference between the 1.0% and 1.5% groups (Table 4). The coefficient variation of initial body length (CVBL$_i$) ($F$ = 1.062, *df* = 5, *p* = 0.427), final body length (CVBL$_f$) ($F$ = 1.829, *df* = 5, *p* = 0.181), and initial body weight (CVBL$_i$) ($F$ = 0.493, *df* = 5, *p* = 0.776) were similar among the treatment groups, whereas the coefficient variation of the final body weight (CVBW$_f$) ($F$ = 7.433, *df* = 5, *p* = 0.002) of the 0.5% group was significantly lower than those of the other groups (Table 4).

**Table 4.** Condition factor, hepatosomatic index, viscerasomatic index, coefficient variation of body length, and body weight of juvenile mandarin fish *Siniperca scherzeri* fed the experimental diets at various feeding rates for 10 weeks *.

| Feeding Rate (% BW$^{-1}$) | CF [1] | HSI (%) [2] | VSI (%) [3] | CVBL$_i$ (%) [4] | CVBL$_f$ (%) [5] | CVBW$_i$ (%) [6] | CVBW$_f$ (%) [7] |
|---|---|---|---|---|---|---|---|
| 0.5 | 0.93 ± 0.01 [a] | 0.90 ± 0.07 [a] | 5.38 ± 0.20 [a] | 7.1 ± 1.01 | 10.4 ± 1.07 | 19.8 ± 0.83 | 35.7 ± 1.98 [b] |
| 1.0 | 1.05 ± 0.00 [b] | 1.34 ± 0.12 [a,b] | 6.28 ± 0.13 [ab] | 7.4 ± 0.53 | 7.8 ± 0.55 | 20.5 ± 1.73 | 21.0 ± 1.41 [a] |
| 1.5 | 1.15 ± 0.01 [c] | 1.66 ± 0.04 [b] | 6.51 ± 0.11 [ab] | 7.1 ± 0.58 | 7.6 ± 0.72 | 18.9 ± 1.24 | 21.6 ± 1.05 [a] |
| 2.0 | 1.14 ± 0.01 [c] | 1.44 ± 0.17 [b] | 6.95 ± 0.47 [b] | 7.9 ± 1.78 | 8.0 ± 0.66 | 23.3 ± 2.27 | 23.8 ± 2.24 [a] |
| 2.5 | 1.16 ± 0.00 [c] | 1.52 ± 0.04 [b] | 7.31 ± 0.24 [b] | 6.4 ± 0.75 | 8.9 ± 0.93 | 19.8 ± 2.79 | 26.1 ± 3.22 [a] |
| 3.0 | 1.20 ± 0.02 [d] | 1.48 ± 0.08 [b] | 7.17 ± 0.22 [b] | 7.6 ± 0.41 | 8.4 ± 0.41 | 21.5 ± 3.30 | 25.7 ± 1.08 [a] |

* Values (mean ± SE of three replicate groups) with different superscripts in the same column are significantly different ($p < 0.05$). Condition factor (CF), hepatosomatic index (HSI), viscerosomatic index (VSI), coefficient variation of body length (CVBL) and coefficient variation of body weight (CVBW). [1] CF = 100 × (weight of fish/length of fish$^3$). [2] HSI (%) = 100 × (weight of liver/weight of fish). [3] VSI (%) = 100 × (weight of viscera/weight of fish)]. [4] CVBL (%)$_i$ = 100 × (standard deviation of initial length of fish/mean initial length of fish). [5] CVBL (%)$_f$ = 100 × (standard deviation of final length of fish/mean final length of fish). [6] CVBW(%)$_i$ = 100 × (standard deviation of initial weight of fish/mean initial weight of fish). [7] CVBW (%)$_f$ = 100 × (standard deviation of final weight of fish/mean final weight of fish).

The moisture, protein, lipid, and ash contents of the fish at the end of the experiment are shown in Table 5. No significant differences occurred in the moisture ($F$ = 1.842, *df* = 5, *p* = 0.179), protein ($F$ = 1.782, *df* = 5, *p* = 0.191), or ash ($F$ = 1.678, *df* = 5, *p* = 0.214) contents among the groups with various feeding rates; however, the lipid ($F$ = 39.589, *df* = 5, *p* < 0.0001) content significantly increased as the feeding rate increased.

**Table 5.** Whole body proximate composition (% of wet weight) of juvenile mandarin fish *Siniperca scherzeri* fed the experimental diets with various feeding rates for 10 weeks *.

| Feeding Rate (% BW$^{-1}$) | Moisture (%) | Crude Protein (%) | Crude Lipid (%) | Ash (%) |
|---|---|---|---|---|
| 0.5 | 73.6 ± 1.67 | 17.5 ± 1.20 | 1.3 ± 0.33 [a] | 5.8 ± 0.07 |
| 1.0 | 71.9 ± 0.52 | 19.1 ± 0.30 | 2.5 ± 0.29 [b] | 5.9 ± 0.25 |
| 1.5 | 71.9 ± 0.90 | 18.3 ± 0.65 | 3.6 ± 0.07 [c] | 5.0 ± 0.44 |
| 2.0 | 71.1 ± 0.48 | 19.1 ± 0.52 | 4.4 ± 0.23 [c,d] | 4.9 ± 0.12 |
| 2.5 | 70.0 ± 1.23 | 19.7 ± 0.85 | 4.4 ± 0.07 [d] | 5.4 ± 0.53 |
| 3.0 | 70.2 ± 0.50 | 20.0 ± 0.10 | 4.3 ± 0.09 [c,d] | 5.2 ± 0.10 |

* Values (mean ± SE of three replicate groups) with different superscripts in the same column are significantly different ($p < 0.05$).

The HCT, GLU, TP, TCHO, GOT, GPT, and HDLC concentrations of the fish blood are summarized in Table 6. The HCT content was significantly different among the groups ($F$ = 4.395, *df* = 5, *p* < 0.017) and was significantly higher in the groups with the 2.5% and 3.0% feeding rate ($p < 0.05$) compared with that of the 0.5%, 1%, and 1.5% groups ($p < 0.05$). The TCHO ($F$ = 9.126, *df* = 5, *p* = 0.001) content was significantly higher in the 1.0–3.0% group than the 0.5% group ($p < 0.05$). The TP ($F$ = 2.351, *df* = 5, *p* = 0.105), GOT ($F$ = 1.124, *df* = 5, *p* = 0.399), GLU ($F$ = 2.108, *df* = 5, *p* = 0.135), GPT ($F$ = 0.885, *df* = 5, *p* = 0.538), and HDLC ($F$ = 1.498, *df* = 5, *p* = 0.262) contents did not significantly differ among any of the groups.

**Table 6.** Hematological response of the blood of juvenile mandarin fish *Siniperca scherzeri* fed the experimental diets with various feeding rates for 10 weeks *.

| Feeding Rate(% BW$^{-1}$) | HCT (%) [1] | GLU (mg dL$^{-1}$) [2] | TP (g dL$^{-1}$) [3] | TCHO (mg dL$^{-1}$) [4] | GOT (U L$^{-1}$) [5] | GPT (U L$^{-1}$) [6] | HDLC (U L$^{-1}$) [7] |
|---|---|---|---|---|---|---|---|
| 0.5 | 41.9 ± 1.0 [a] | 167.8 ± 23.5 | 4.0 ± 0.1 | 131.2 ± 20.8 [a] | 92.2 ± 28.7 | 19.8 ± 6.1 | 102.9 ± 3.3 |
| 1.0 | 44.6 ± 1.5 [a,b] | 234.3 ± 33.2 | 4.6 ± 0.1 | 201.1 ± 1.9 [b] | 102.3 ± 14.7 | 16.6 ± 1.9 | 110.0 ± 0.1 |
| 1.5 | 46.1 ± 1.5 [a,b] | 224.5 ± 31.3 | 5.2 ± 0.2 | 231.7 ± 15.7 [b] | 90.1 ± 26.0 | 25.8 ± 3.5 | 116.4 ± 6.4 |
| 2.0 | 46.3 ± 1.0 [a,b] | 271.6 ± 15.8 | 4.7 ± 0.1 | 209.7 ± 4.5 [b] | 51.5 ± 5.0 | 19.2 ± 1.6 | 110.0 ± 0.1 |
| 2.5 | 47.9 ± 0.6 [b] | 312.1 ± 64.5 | 4.4 ± 0.5 | 204.3 ± 9.6 [b] | 67.3 ± 17.3 | 30.3 ± 18.4 | 105.8 ± 2.1 |
| 3.0 | 49.2 ± 0.2 [b] | 276.9 ± 13.0 | 4.5 ± 0.2 | 214.6 ± 3.3 [b] | 65.4 ± 2.8 | 8.7 ± 1.2 | 116.3 ± 7.9 |

* Values (mean ± SE of three replicate groups) with different superscripts in the same row are significantly different ($p < 0.05$).
[1] HCT = Haematocrit. [2] GLU = Glucose. [3] TP = Total protein. [4] TCHO = Total cholesterol. [5] GOT = glutamic oxaloacetic transaminase. [6] GPT = glutamic pyruvic transaminase. [7] HDLC = high density lipoprotein cholesterol.

The quadratic broken-line model based on the WG and SGR, and the quadratic model (second-order polynomial) analysis based on the FE of juvenile *S. scherzeri* allocated to the different feeding rate groups estimated the optimum feeding rate to be 1.88–2.80% BW day$^{-1}$ (Figure 1).

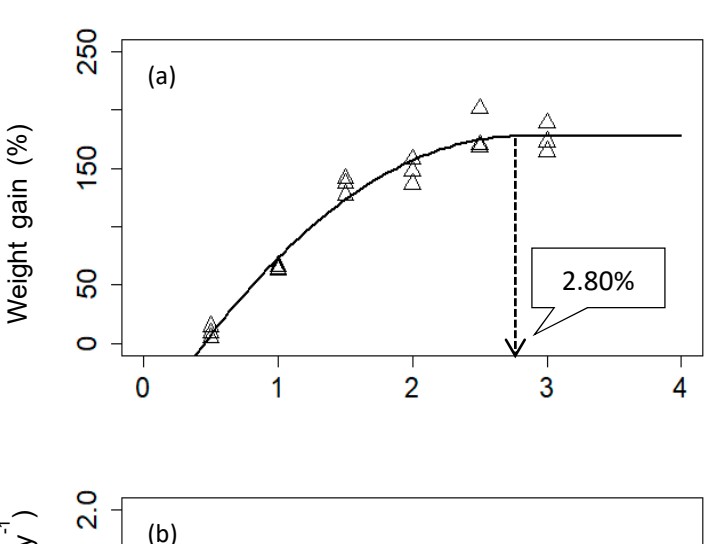

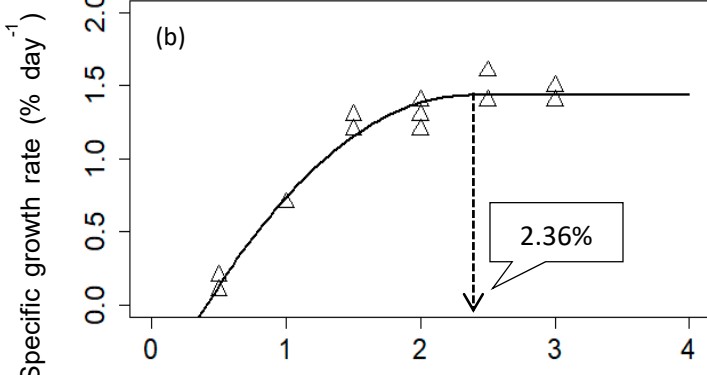

**Figure 1.** *Cont.*

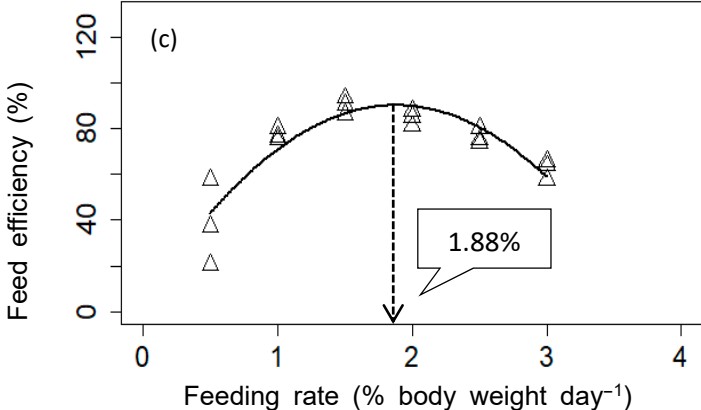

**Figure 1.** Relationship between feeding rate and weight gain (**a**), specific growth rate (**b**) and feed efficiency (**c**). The optimum feeding rate for juvenile mandarin fish *Siniperca scherzeri* was estimated based on the quadratic broken-line regression analysis for weight gain (%) and specific growth rate (%), and on the quadratic (second-order polynomial) regression analysis for feed efficiency (%) against feeding rate (%), respectively.

## 4. Discussion

Increasing the feeding rate had a significant influence on the growth of juvenile *S. scherzeri*, which is a result similar to findings of studies on other species [14,19,24–33]. The WG and SGR also significantly increased as the feeding rate increased. The groups fed at a rate of 2.5% and 3.0% BW day$^{-1}$ exhibited the greatest increase in WG and SGR, and there was no significant difference between these two groups (Table 2). This is in line with the previous findings showing that the growth rate increases as the feeding rate increases [19,25,26,29,30,33], and that increasing the feed supply above a certain amount results in no increase, or even a decrease, in the growth rate [14,24,27,32].

Mandarin fish are highly carnivorous, with an ecological habit of directly eating small live fish. The development of mandarin fish aquaculture technology is currently at a rudimentary stage, and the most difficult problem has been the development of suitable feed for this species. In other words, information on various nutrient requirements such as optimal protein content is insufficient. In the recent research, it was reported that a high content of fish meal is required as a dietary source to meet the optimal dietary protein and lipid content [9], and few studies have reported on the substitute of fishery derivatives or the use of plant protein sources. Therefore, the 76% fishmeal content and the composition of other ingredients in the experimental diet used in the previous study was applied in this study [9]. A study on the growth performance of mandarin fish using other materials as fishmeal substitutes should be completed in the future to sustain the aquaculture industry.

The tempo of growth in relation to the feeding rate usually varies with species and body size. For example, the optimal feeding rates for the grass carp *Ctenopharyngodon idella* [28] and the tropical bagrid catfish *Mystus nemurus* are 1.97% BW day$^{-1}$ and 2.5% BW day$^{-1}$, respectively [14], while that of the Chinese sucker *Myxocyprinus asiaticus* is 3.1% BW day$^{-1}$ [31]. The optimal feeding rate for the European sea bass *Dicentrarchus labrax* was reported to be 3.0–3.5% BW day$^{-1}$ for a 2.6 g fish [29], and that of the Indian major carp *Catla catla* was 3.81–4.19% BW day$^{-1}$ [24]. The optimal feeding rates for the green sturgeon *Acipenser medirostris* were 5.3% BW day$^{-1}$ at 7.5 g, 5.7% BW day$^{-1}$ at 2.6 g, and 7.1% BW day$^{-1}$ at 1.6 g [32], and those of the olive flounder *Paralichthys olivaceus* were 5.1% BW day$^{-1}$ (5 g) and 3.4% BW day$^{-1}$ (20 g) [33].

The present findings indicate that the feeding rate is positively related to the FE, PER, and PR, as the three indicators were at the highest levels when the feeding rate was 1.5%. Whereas, at feeding rates over 1.5%, they decreased. While many studies of fish in different taxa reported that the FE increased until the feeding rate reached a certain level, the FE, PER, and PR decreased when the feeding rate went above the threshold [14,27,28,30,31,33,34].

The effects of the feeding rate on the growth rate are closely associated with the digestion and absorption of the feed by the fish, and a higher feeding rate reduces digestion efficiency [35]. If excessive feed is provided, too much time is required for the feed to pass through the digestive tract, which prevents efficient digestion [36–38]. Therefore, it is crucial to provide less feed than needed for satiation, while ensuring sufficient supplies for growth, to reduce feeding costs and water pollution and to increase management convenience and feed availability [25,39,40]. However, contrasting results have been observed for certain species, as there have been reports of no differences in feeding efficiency, protein conversion rate, or protein content with changes in the feeding rate [25].

Our findings are consistent with previous studies on several species [12,41,42] in that, except for the 0.5% rate, the feeding rate did not affect the CVBL and CVBW. This suggests that any feeding rate in the range of 1.0–3.0% BW day$^{-1}$ would ensure the growth of juvenile *S. scherzeri* (18–54 g), regardless of weight and length. In addition, the feeding rate had an impact on the CF, HSI, and VSI, which is in agreement with previous reports [14,19,27,43–45]. We believe that the HSI and VSI increased with increasing feeding rates because they are used as energy storage for the liver and intestines [28,44]. However, a few studies in other species (e.g., [25,26]) have reported that the feeding rate has no impact on CF, HSI, and VSI.

The crude lipid content significantly increased with increasing feeding rates because the energy from the extra feed was stored as body fat [2,14,19,24,27–29,31,33,44,46,47]. However, some studies have reported that the feeding rate has no effect on lipid content [25,26,34]. Interestingly, our study showed that the increase in feeding rate did not influence the blood contents of GLU, GPT, GOT, TP, and HDLC. Instead, the concentrations of HCT and TCHO increased. This was probably because increasing the rate of feeding reduces stressful conditions, but it is still unclear what the mechanism is. A similar trend was found in the common carp (*Cryprinus carpio*, [30]), olive flounder (*Paralichthys olivaceus*, [33]), white sturgeon (*Acipenser transmontanus*, [44]), and the Brazilian sardine (*Sardinella brasiliensis*, [36]). In contrast, Cho et al. [25] reported that the feeding rate did not affect the serum concentrations of glucose, total protein, triglyceride, or GPT in the olive flounder because fish WG decreased proportionally with the reduction in feed consumption. Similarly, reducing the feeding rate caused stress and, consequently, increased the serum concentration of GOT in some fish [33,38].

Based on the results of the regression analysis (Figure 1), the optimal feeding rate should be considered in terms of both growth (i.e., WG and SGR) and FE. In previous studies [29,47,48], FE decreased in a curve with an increasing feeding rate, which was consistent with the results of this study. In this study, the highest FE was found at 1.5% BW day$^{-1}$, whereas the fish with the highest growth rate fed at satiation level (i.e., 2.5–3.0% BW day$^{-1}$) had an FE of only 62.4–76.3%. A similar phenomenon was observed in the same species [49], European sea bass [29] raised in seawater and freshwater, gilthead sea bream [48], rainbow trout [50], and yellowtail flounder [51]. It might be due to the retention of low energy and protein in the body in relation to the high energy requirement for feed utilization at a high feeding rate [52]. Therefore, this information will be useful for fish farmers for this species to determine the optimal feeding rate in terms of high growth rate or economical FE.

Our findings clearly showed that the growth rate, body composition, and blood properties of juvenile *S. scherzeri* were positively related to the feeding rate up to an optimal level, whereas above the optimum feeding rate, these outcomes were negatively affected. The regression analysis based on the WG, SGR, and FE for the 0.5–3.0% BW day$^{-1}$ feeding rate estimated the optimum feeding rate to be 1.88–2.80% BW day$^{-1}$ for *S. scherzeri* weighing between 18 g and 55 g under 26.9 °C condition. This represents key information that could help farmers of *S. scherzeri* to optimize their feeding systems. Further studies on the requirements and utilization of nutrients of *S. scherzeri* fingerlings are needed to formulate optimal feeding regimes that can promote better growth.

**Author Contributions:** Conceptualization, Y.-O.K. and S.-Y.O.; methodology, Y.-O.K. and S.-Y.O.; software, Y.-O.K. and S.-Y.O.; validation, S.-Y.O. and T.K.; formal analysis, S.-Y.O. and T.K.; resources, Y.-O.K.; data curation, Y.-O.K. and S.-Y.O.; writing—original draft preparation, Y.-O.K., S.-Y.O. and T.K.; writing—review and editing, S.-Y.O. and T.K.; visualization, S.-Y.O. and T.K.; supervision, S.-Y.O. and T.K.; project administration, T.K.; funding acquisition, T.K. All authors have read and agreed to the published version of the manuscript.

**Funding:** This research was funded by the Ministry of Oceans and Fisheries, Korea (grant number 20210642, "Development of 3-D Ocean Current Observation Technology for Efficient Response to Maritime Distress").

**Institutional Review Board Statement:** This study was conducted according to the guidelines of the project "Development of diet and acclimation technology in the mandarin fish *Siniperca scherzeri*" approved by Chungcheongbuk-do Inland Fisheries Research Institute, Korea (R&D/BSPM60410-1166-3).

**Informed Consent Statement:** Not applicable.

**Data Availability Statement:** The datasets generated and/or analyzed during the current study are available from the corresponding author on reasonable request.

**Conflicts of Interest:** The authors declare no conflict of interest.

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
