# Peer review of "Effects of the Feeding Rate on Growth Performance, Body Composition, and Hematological Properties of Juvenile Mandarin Fish Siniperca scherzeri in a Recirculating Aquaculture System"

_sustainability, doi:10.3390/su13158257_

Round 1

Reviewer 1 Report

This piece of research evaluates the effect of different feeding rates on growth performance, chemical composition and blood biochemistry in juvenile Siniperca scherzeri. The results are of interest in the aquaculture of this species. However, the manuscript must be improved for increasing the value of this contribution. There are several concerns related to animal welfare that authors should address, sustainability of the ingredient used for feed formulation, also blood analysis methodology must be improved, and some results are missing.

Authors should consider the following specific comments:

Title: Use “serum biochemistry” instead of “blood parameters”

Line 19. Specify lipid content of experimental feed too.

Line 28. In the present study, the optimal feeding rate is estimated at 2.37% but only for 26.9 ºC. Optimum values can vary in function of water temperature so values should set up for each specific water temperature conditions. Indeed, the feeding rates should adjusted depending the water conditions. This point should be highlighted in the conclusions of the study.

Line 28. Optimal rate was estimated in function of weight gain, but SGR and Feed efficiency are also key criteria for choosing the optimal feeding rate. These days the use of feed together growth performance should be considered for selecting the adequate feeding rates for optimal growth and feed utilisation efficiency. Is the same results obtained if SGR and feed efficiency values are used in the broke-line analysis? Authors should consider also those other parameters for stablishing the optimal feeding

Line 55. Use “Blood properties” is not adequate, it is more adequate refer to “serum/serum biochemistry”

Line 81. Check “total 1-day”

Line 90. “Fish meal from anchoy Engraulis japonicas was”

Line 91. Provide detail about the equipment used for making the pellets and the procedure. Was the diet extruded?

Table 1. The experimental diet was formulated using 76% fishmeal, these days aquaculture production should be sustainable and should avoid an excessive use of fishery derivatives. One of the main concerns about the formulation of the experimental diet is the high proportion of fishmeal used. This feed formulation might be considered no sustainable regarding the use of natural resources owing to it is highly dependent of wild fisheries. Authors must explain and justify the reasons for using this high proportion of fishmeal in their feed formulation.

Lines 114 and 128. The way authors explain is confusing owing to it seems that 150 mg/L 2-phenoxyethanol is used for both anesthetizing and sacrificing fish. Authors should explain better the procedure for sacrificing.

Line 127. Were the 15 blood samples pooled or analysed individually? Specify

Line 129. Was blood samples withdraw before or after killing fish? Specify

Line 132-134. Authors must explain the procedures used for each determination. The way experimental procedures are explained do not allow replication by other researchers.

Table 2. What is the sense of feeding fish 0.5 feeding rate? The main concern about this treatment is related to animal welfare. Clearly, fish were underfed for 10 weeks so which is the scientific value of these data? From my point of view this treatment should be omitted.

Table 3. The way authors estimated feed efficiency seems unusual. Please consider for the calculation the formula detailed in Glencross et al. 2007. A feed is only as good as its ingredients – a review of ingredient evaluation strategies for aquaculture feeds. Aquaculture Nutrition 13; 17–34

Tables 2, 4. Omit the superscript “ns” in column without statistical differences.

Table 6 is missing in the manuscript so it is not possible to revise results of blood biochemistry.

Fig 1. What about the relationship between feeding rate and SGR and FE?

Line 287. An excessive feeding rate reduces stressful conditions; by probably 0.5% feeding rate compromises the animal welfare in this trial. These days there is noticeable awareness in the international scientific community about this aspect.

Author Response

Reviewer 1

This piece of research evaluates the effect of different feeding rates on growth performance, chemical composition and blood biochemistry in juvenile Siniperca scherzeri. The results are of interest in the aquaculture of this species. However, the manuscript must be improved for increasing the value of this contribution. There are several concerns related to animal welfare that authors should address, sustainability of the ingredient used for feed formulation, also blood analysis methodology must be improved, and some results are missing.

Authors should consider the following specific comments:

Title: Use “serum biochemistry” instead of “blood parameters”

Answer: Most of the blood properties in this study are serum parameters, but hematocrit must be measured with whole blood. Therefore, it is expressed as hematological properties to represent both serum and whole blood parameters.

Line 19. Specify lipid content of experimental feed too.

Answer: Following the reviewer’s suggestion, we have inserted the lipid content with protein content as follows: (55.4% protein, 14.1% lipid).

Line 28. In the present study, the optimal feeding rate is estimated at 2.37% but only for 26.9 ºC. Optimum values can vary in function of water temperature so values should set up for each specific water temperature conditions. Indeed, the feeding rates should adjusted depending the water conditions. This point should be highlighted in the conclusions of the study.

Answer: Following the reviewer’s suggestion, we have revised the text in the Abstract and in the end of Discussion. We have revised “at 2.37% BW day-1” to “at 1.88- 2.8% BW day-1 depending on weight gain, specific growth rate, and feed efficiency under 26.9 ºC condition” in Abstract and in end of Discussion.  

Line 28. Optimal rate was estimated in function of weight gain, but SGR and Feed efficiency are also key criteria for choosing the optimal feeding rate. These days the use of feed together growth performance should be considered for selecting the adequate feeding rates for optimal growth and feed utilisation efficiency. Is the same results obtained if SGR and feed efficiency values are used in the broke-line analysis? Authors should consider also those other parameters for stablishing the optimal feeding

Answer: Following the reviewer’s suggestion, we have inserted the optimal feeding rate (see Fig. 1) through estimations of function with weight gain, specific growth rate and feed efficiency, respectively. The optimal feeding rate for juvenile mandarin fish was calculated based on Quadratic broken-line regression analysis of weight and specific growth rate against feeding rate, and on the Quadratic model (second-order polynomial) analysis against feeding rate by method of Lee et al.(2014) (reference n (46). And we considered the results according to the weight gain, specific growth rate, and feed efficiency in the Discussion. 

Line 55. Use “Blood properties” is not adequate, it is more adequate refer to “serum/serum biochemistry”

Answer: As mentioned above, hematocrit which is whole blood analysis parameter is included in this study, so it is expressed as hematological properties. 

Line 81. Check “total 1-day”

Answer: Following the reviewer’s comment, we have revised the text to clarify the meaning of feed amount of each day, “The total feed amount of each day was divided into two equal portions, which were given to the fish at 09:00 and 17:00 each day.”

Line 90. “Fish meal from anchoy Engraulis japonicas was”

Answer: Following the reviewer’s comment, we have revised “The anchovy Engraulis japonicus” to “The anchovy fish meal”.

Line 91. Provide detail about the equipment used for making the pellets and the procedure. Was the diet extruded?

Answer: Following the reviewer’s comment, we have inserted the detailed explanation about preparing the experimental diet as follows;

For preparing the experimental diet, all the dry ingredients were finely ground in a laboratory mill (HKPM-100, Hankook Power System, Seoul, South Korea) and well-mixed by mixer (NVM-18, Daeyung Bakery Machinery Co. Ltd., Seoul, South Korea). Squid liver oil, soybean oil and double-distilled water were then added into the mixture and mixed. The well-mixed final moist mash were pelleted through a meat chopper machine (SMC-32, SL Co., Incheon, South Korea) adjusted with a 3mm diameter. The produced pellets were dried at room temperature overnight, and stored at -30ºC until used.

Table 1. The experimental diet was formulated using 76% fishmeal, these days aquaculture production should be sustainable and should avoid an excessive use of fishery derivatives. One of the main concerns about the formulation of the experimental diet is the high proportion of fishmeal used. This feed formulation might be considered no sustainable regarding the use of natural resources owing to it is highly dependent of wild fisheries. Authors must explain and justify the reasons for using this high proportion of fishmeal in their feed formulation.

Answer: Mandarin fish are highly carnivorous fish, showing an ecological habit of directly eating small live fish. The development of mandarin fish aquaculture technology is currently at a rudimentary stage, and the most difficult problem has been the development of suitable feed for this species. In other words, information on various nutrient requirements such as optimal protein content is insufficient.  

In the recent research, limited amount of studies have been reported on the optimal dietary protein and lipid content (reference 9) required for this species using high content of fish meal as dietary sourcesand few studies have been reported on the replacement of fishery derivatives or use of plant protein sources. Therefore, the 76% fishmeal content and composition of other ingredients in the experimental diet applied in this study followed the previous study (reference 9). As suggested by the reviewer, a study on the growth performance of mandarin fish using other materials as fishmeal substitutes should be done in the future to sustain aquaculture industry. We inserted this argument in the Discussion.

Lines 114 and 128. The way authors explain is confusing owing to it seems that 150 mg/L 2-phenoxyethanol is used for both anesthetizing and sacrificing fish. Authors should explain better the procedure for sacrificing.

Answer: Following the reviewer’s comment, we have revised the text to clarify the meaning of sacrificing fish (Line 116-117), “At the end of the experiment, 9 fish (3 fish per tank) in each treatment group were randomly sacrificed by anesthetizing in 150 mg/L 2-phenoxyethanol solution for 1 minute, and stored in a -40ºC freezer until the analysis of body composition.”

Line 127. Were the 15 blood samples pooled or analysed individually? Specify

Answer: Following the reviewer’s comment, we have revised the text to clarify the meaning of analyzing blood samples of fish, “Before sacrificing fish, blood sample of each fish was individually collected from the tail artery using a heparinized syringe.

Line 129. Was blood samples withdraw before or after killing fish? Specify

Answer: Following the reviewer’s comment, we have revised the text to clarify the meaning of analyzing blood samples of fish, “Before sacrificing fish, blood sample of each fish was collected from the tail artery using a heparinized syringe, and analyzed individually.

Line 132-134. Authors must explain the procedures used for each determination. The way experimental procedures are explained do not allow replication by other researchers.

Answer: The method of blood sampling and serum collection for blood analysis are already presented in Materials and Methods. Each blood property was measured according to the manual of the presented analytical instrument (i.e., DRI-CHEM NX500i). Therefore, other researchers can refer to the manual for analysis of the instrument in order to measure same blood properties of this study.

Table 2. What is the sense of feeding fish 0.5 feeding rate? The main concern about this treatment is related to animal welfare. Clearly, fish were underfed for 10 weeks so which is the scientific value of these data? From my point of view this treatment should be omitted.

Answer: Commercial aquaculture farmers intentionally or non-intentionally choose fasting (i.e. 0% feeding rate) or limited feeding (i.e. 0.5% feeding rate of this study) for prolonged periods to save feed, enhance the growth rate, and decrease mortality caused by environmental deterioration, diseases and handling for selection or transport. Although the 0.5% feeding rate treatment was insufficient for the growth of juvenile mandarin fish in this experiment, no mortality of this species due to the low feeding rate occurred, showing a weight gain of about 9.7% and an SGR of 0.13%. In other words, there was no result affecting animal welfare such as mortality in the 0.5% feeding rate treatment during the experimental period. We also believe that these results are valuable in terms of limited feeding strategies, such as fasting or 0.5% feeding rate, which occur for various reasons in aquaculture sites.

Table 3. The way authors estimated feed efficiency seems unusual. Please consider for the calculation the formula detailed in Glencross et al. 2007. A feed is only as good as its ingredients – a review of ingredient evaluation strategies for aquaculture feeds. Aquaculture Nutrition 13; 17–34

Answer: The calculation formula of feed efficiency presented in this study is not different from that of the literature presented by reviewer. The difference is that our suggested feed efficiency is expressed as a percentage, and is used in many literatures as follows:

Verbeeten, B.E., Carter, C.G., Purser G.J. 1999. The combined effect of feeding time and ration on growth performance and nitrogen metabolism of greenback flounder. Journal of Fish Biology 55, 1328-1343.

Eroldgan, O.T., Kumlu M. Aktas, M. 2004. Optimum feeding rates for European sea bass Dicentrarchus labrax L. reared in seawater and freshwater. Aquaculture 231, 501-515. 

Mizaur, R.M., Park, G., Yun, H., Lee, S., Choi, S., Bai, S.C. 2014. The effects of feeding rates in juvenile Korean rockfish (Sebastes schlegeli) reared at 17℃ and 20℃ water temperatures. Aquaculture international 22, 1121-1130.

Bu, X., Lian, X., Zhang, Y., Yang, C., Cui, C., Che, J., Tang, B., Su, B., Zhou, Q., Yang, Y. 2017. Effects of feeding rates on growth, feed utilization, and body composition of juvenile Pseudobagrus ussuriensis. Aquaculture International 25, 1821-1831.

Sankian, Z., Khosravi. S., Kim, Y.I., Lee, S.M. 2018. Effects of dietary inclusion of yellow mealworm (Tenebrio molitor) meal on growth performance, feed utilization, body composition, plasma biochemical indices, selected immune parameters and antioxidant enzyme activities of mandarin fish (Siniperca scherzeri) juveniles. Aquaculture 496, 79-87. 

Tables 2, 4. Omit the superscript “ns” in column without statistical differences.

Answer: Corrected.

Table 6 is missing in the manuscript so it is not possible to revise results of blood biochemistry.

Answer: It’s our mistake. We inserted Table 6 in MS.

Fig 1. What about the relationship between feeding rate and SGR and FE?

Answer: Following the reviewer’s suggestion, we have inserted the optimal feeding rate (see Fig. 1) through estimations of function with weight gain, specific growth rate and feed efficiency, respectively. The optimal feeding rate for juvenile mandarin fish was calculated based on Quadratic broken-line regression analysis of weight and specific growth rate against feeding rate, and on the Quadratic model (second-order polynomial analysis) against feeding rate by method of Lee et al.(2014) (reference n (46). And we considered the results according to the weight gain, specific growth rate, and feed efficiency in the Discussion. 

Line 287. An excessive feeding rate reduces stressful conditions; by probably 0.5% feeding rate compromises the animal welfare in this trial. These days there is noticeable awareness in the international scientific community about this aspect.

Answer: As mentioned earlier, commercial aquaculture farmers intentionally or non-intentionally choose fasting (i.e. 0% feeding rate) or limited feeding (i.e. 0.5% feeding rate of this study) for prolonged periods to save feed, enhance the growth rate, and decrease mortality caused by environmental deterioration, diseases and handling for selection or transport. Although the 0.5% feeding rate treatment was insufficient for the growth of juvenile mandarin fish in this experiment, no mortality of this species due to the low feeding rate occurred, showing a weight gain of about 9.7% and an SGR of 0.13%. In other words, there was no result affecting animal welfare such as mortality in the 0.5% feeding rate treatment during the experimental period. We also believe that these results are valuable in terms of limited feeding strategies, such as fasting or 0.5% feeding rate, which occur for various reasons in aquaculture sites.

Reviewer 2 Report

General comments:

The paper by Yi-Oh Kim and colleagues can represents a good starting point as reference data for future studies about this topic. Moreover, implementing the quality of foraging of farming species could certainly be a valid tool in aquaculture. Despite the present study has a good experimental idea, it needs revisions both for contents and structural organization in order to improve the quality of the manuscript. With particular care, I suggest the authors to review the materials and methods dedicated section. I would also consider a language revision; there are in fact long periods and several repetitions.

Introduction:

Introduction section is well presented and supported with updated citations. However, I suggest the authors not to use the term “water pollution” (Lines 45-48), as it is not a real water pollution but an increase in organic substances that is disposed of over time.

 Materials and Methods;

Materials and Methods section needs revisions:

  • Lines 64-65: It would be advisable for the authors to report in the text all the types and origin data of the instrumentation used (filters, tanks, etc.).
  • Line 117: Which procedure was used to sacrifice the samples? Please add these informations.

Results;

Results are well presented and valid.

 Discussion;

In the discussion section, the authors support their results with updated bibliography, and then does not need furthering corrections. 

Caption;

Table 1: I suggest a revision of the layout. Moreover, the percentages reported are not clear. 

Author Response

Reviewer 2

General comments:

The paper by Yi-Oh Kim and colleagues can represents a good starting point as reference data for future studies about this topic. Moreover, implementing the quality of foraging of farming species could certainly be a valid tool in aquaculture. Despite the present study has a good experimental idea, it needs revisions both for contents and structural organization in order to improve the quality of the manuscript. With particular care, I suggest the authors to review the materials and methods dedicated section. I would also consider a language revision; there are in fact long periods and several repetitions.

Introduction:

Introduction section is well presented and supported with updated citations. However, I suggest the authors not to use the term “water pollution” (Lines 45-48), as it is not a real water pollution but an increase in organic substances that is disposed of over time.

Answer: Corrected.

Materials and Methods;

Materials and Methods section needs revisions:

Lines 64-65: It would be advisable for the authors to report in the text all the types and origin data of the instrumentation used (filters, tanks, etc.).

Answer: The small recirculation culture system including nitrification filtration tank, foam separator and circular fiberglass water tank used in this study was not purchased, but manufactured. We inserted the additional information of experimental system, “a submerged nitrification filter (3000 L, specific surface area of filter media 200 m2 m3-1), foam separator (100L, retention time 90 seconds), and 18 circular fiberglass culture tanks (300 L, flow rate 30 L hour-1).  

Line 117: Which procedure was used to sacrifice the samples? Please add these informations.

Answer: Following the reviewer’s suggestion, we have revised the text to clarify the meaning of sacrificing of fish (Line 116-117), “At the end of the experiment, 9 fish (3 fish per tank) in each treatment group were randomly sacrificed by anesthetizing in 150 mg/L 2-phenoxyethanol solution for 1 minute, and stored in a -40ºC freezer until the analysis of body composition.”

Results;

Results are well presented and valid.

Discussion;

In the discussion section, the authors support their results with updated bibliography, and then does not need furthering corrections.

Caption;

Table 1: I suggest a revision of the layout. Moreover, the percentages reported are not clear. 

Answer: Corrected.

Reviewer 3 Report

In the present study, authors aimed at identifying the optimal fedding rate for mandarin fish juveniles between 18 and 54 g.

In general, the experiment is well conceived, the number of individuals and replicates is scientifically sound and the results section is well described. However, the topic, in my opinion, does not fit with the journal: the objective of this experiment is the optimization of fish growth parameters, not sustainability of fish farming. Of course, optimize the feeding rate, avoiding overfeeding, concerns sustainability, but this latter has a broader mean, that is not adequately detailed in the introduction.

Moreover, the experimental diet used in this experiment has a very high percentage of fish meal and oil, absolutely not sustainable in terms of ingredients composition. I suggest another journal of the MDPI group, like FISHES.

I have also some suggestions to improve the manuscript quality:

  • The size range considered by authors is very wide for the juvenile stage and an optimal feeding rate cannot be estimated for a 25 g growth range. In the juvenile stage, fish growth is very rapid and feeding rate should be calculated for smaller growth ranges (few g). I suggest to have a look to reference tables of commercial aquafeeds for juveniles.
  • Authors did not specified the temperature. I guess it is constant in a closed system, however it must be indicated, as growth parameters vary according to temperature and are specific for a specific range of temperature.
  • Authors, in their model, considered only SGR, but not FCR. When searching for an optimal feeding rate, feed conversion rate should be considered, in particular when one of the objective is to increase sustainability.

Author Response

Reviewer 3

In the present study, authors aimed at identifying the optimal fedding rate for mandarin fish juveniles between 18 and 54 g. In general, the experiment is well conceived, the number of individuals and replicates is scientifically sound and the results section is well described. However, the topic, in my opinion, does not fit with the journal: the objective of this experiment is the optimization of fish growth parameters, not sustainability of fish farming. Of course, optimize the feeding rate, avoiding overfeeding, concerns sustainability, but this latter has a broader mean, that is not adequately detailed in the introduction. Moreover, the experimental diet used in this experiment has a very high percentage of fish meal and oil, absolutely not sustainable in terms of ingredients composition. I suggest another journal of the MDPI group, like FISHES.

Answer: Mandarin fish is a promising aquaculture candidate species in regions such as China, Korea, and Vietnam for economic reasons (i.e., high price and good taste) and stock reduction (i.e., overfishing and habitat destruction). However, most commercial production of this species is done by live feed due to its high carnivorous nature. For stable and continuous aquaculture of this species, it is necessary to develop a formulated feed and a suitable feeding regime. Although there have been limited recent reports of information on nutrient requirements and feeding regime, few studies have been reported on the replacement of fishery derivatives or use of plant protein sources. The 76% fishmeal content and composition of other ingredients in the experimental diet applied in this study was cited by the previous study (i.e., reference 9). As suggested by the reviewer, a study on the growth performance of mandarin fish using other materials as fishmeal substitutes should be done in the future to sustain aquaculture industry.

I have also some suggestions to improve the manuscript quality:

The size range considered by authors is very wide for the juvenile stage and an optimal feeding rate cannot be estimated for a 25 g growth range. In the juvenile stage, fish growth is very rapid and feeding rate should be calculated for smaller growth ranges (few g). I suggest to have a look to reference tables of commercial aquafeeds for juveniles.

Answer: As the reviewer points out, the optimal feeding rate depends on many conditions, including species, size, temperature, and feed composition, etc. Therefore, the optimal feeding rate in this study is specified as the result derived from the limited fish size range and water temperature conditions in which the experiment was conducted. These experimental conditions in this study may serve as a reference for other researchers to conduct additional experiments under other conditions in the future. 

Authors did not specified the temperature. I guess it is constant in a closed system, however it must be indicated, as growth parameters vary according to temperature and are specific for a specific range of temperature.

Answer: Following the reviewer’s suggestion, we have revised the text in the Abstract and in the end of Discussion. We have revised “at 2.37% BW day-1” to “at 1.88- 2.8% BW day-1 depending on weight gain, specific growth rate, and feed efficiency under 26.9 ºC condition” in Abstract and in end of Discussion.    

Authors, in their model, considered only SGR, but not FCR. When searching for an optimal feeding rate, feed conversion rate should be considered, in particular when one of the objective is to increase sustainability.

Answer: Following the reviewer’s suggestion, we have inserted the optimal feeding rate (see Fig. 1) through estimations of function with weight gain, specific growth rate and feed efficiency, respectively. The optimal feeding rate for juvenile mandarin fish was calculated based on Quadratic broken-line regression analysis of weight and specific growth rate against feeding rate, and on the Quadratic model (second-order polynomial) analysis against feeding rate by method of Lee et al.(2014) (reference 22). And we considered the results according to the weight gain, specific growth rate, and feed efficiency in the Discussion. 

Reviewer 4 Report

Review of the manuscript: ‘Effects of feeding rate on growth, body composition, and blood properties of juvenile mandarin fish Siniperca scherzeri in a water reuse system’. The results of this work are instructive to rational development of RAS in aquaculture.

The first thing to point out is that the title of this article does not attract readers’ interest. I found that the author has studied many indicators in this research, but not all indicators must appear in the title. The author should condense a better title to reflect the highlights of this article. water reuse system is better to be replaced by recirculating aquaculture system.

The abstract section should briefly introduce the research background and research significance and clarify the research methods, then introduce the main research results, and finally, give the corresponding conclusions. The abstract of this article looks ok, but some descriptions are redundant, and I hope to make a comprehensive modification.

The material and method section need to be re-integrated and checked to avoid repetition and confuse. All experimental protocols should be better explained.

The number of samples used for biochemical testing is not clear.

Authors mentioned that fish ad libtum which is not correct term in case of fish feeding. May be you mean apparent satiation. Line 68.

Have they been stunned and killed before freezing at -40 °C? This needs to be clearly stated.

Table 6 is missing???

There are aspects of the technical presentation that require attention:

Make sure that all works cited in the text are in the reference list, that the presentation is consistent and that correct information is given.

Define and explain all acronyms and abbreviations on first mention in the text.

On first mention of a species in the text give both the common (trivial) and formal name, and make sure that the presentation is correct and consistent.

Make sure that symbols, sub- and super-scripts, upper- and lower-case are presented correctly, and that there is correct and consistent use of italics, brackets and punctuation etc.

There are mistakes in the reference list, including incorrect reporting, inconsistent presentation, spelling mistakes and problems with use of punctuation etc.

Author Response

Reviewer 4

Review of the manuscript: ‘Effects of feeding rate on growth, body composition, and blood properties of juvenile mandarin fish Siniperca scherzeri in a water reuse system’. The results of this work are instructive to rational development of RAS in aquaculture. The first thing to point out is that the title of this article does not attract readers’ interest. I found that the author has studied many indicators in this research, but not all indicators must appear in the title. The author should condense a better title to reflect the highlights of this article. water reuse system is better to be replaced by recirculating aquaculture system.

Answer: Thank you for your suggestion. We revised the title as “Effects of feeding rate on growth performance, body composition, and hematological properties of juvenile mandarin fish Siniperca scherzeri in a recirculating aquaculture system.”

The abstract section should briefly introduce the research background and research significance and clarify the research methods, then introduce the main research results, and finally, give the corresponding conclusions. The abstract of this article looks ok, but some descriptions are redundant, and I hope to make a comprehensive modification.

Answer: We have edited the abstract to clarify the whole story of research.

The material and method section need to be re-integrated and checked to avoid repetition and confuse. All experimental protocols should be better explained.

The number of samples used for biochemical testing is not clear.

Answer: Corrected

Authors mentioned that fish ad libtum which is not correct term in case of fish feeding.

May be you mean apparent satiation. Line 68.

Answer: Following the reviewer’s suggestion, we have revised “ad libtum” to “apparent satiation”.

Have they been stunned and killed before freezing at -40 °C? This needs to be clearly stated.

Answer: Following the reviewer’s suggestion, we have revised the text to clarify the meaning of sacrificing of fish (Line 116-117), “At the end of the experiment, 9 fish (3 fish per tank) in each treatment group were randomly sacrificed by anesthetizing in 150 mg/L 2-phenoxyethanol solution for 1 minute, and stored in a -40ºC freezer until the analysis of body composition.”

Table 6 is missing???

Answer: It’s our mistake. We inserted Table 6 in MS.

There are aspects of the technical presentation that require attention:

Make sure that all works cited in the text are in the reference list, that the presentation is consistent and that correct information is given.

Answer: Following review’s suggestion, we checked and confirmed the information is given in MS.

Define and explain all acronyms and abbreviations on first mention in the text.

On first mention of a species in the text give both the common (trivial) and formal name, and make sure that the presentation is correct and consistent.

Make sure that symbols, sub- and super-scripts, upper- and lower-case are presented correctly, and that there is correct and consistent use of italics, brackets and punctuation etc.

Answer: Following review’s suggestion, we checked and revised all acronyms and abbreviatons, symbols, sub- and super-scripts, upper- and lower- case etc.

There are mistakes in the reference list, including incorrect reporting, inconsistent presentation, spelling mistakes and problems with use of punctuation etc.

Answer: Following review’s suggestion, we checked and revised the reference list.

Round 2

Reviewer 1 Report

The manuscript has been improved taking into account the reviewers' comments. Authors provide convincing explanations to each of the comments and make adequate modifications in the manuscript.

Just only a couple of minor changes for considering:

lines 131. Units must be "mg L-1"

lines 122 and 117. Unist must be "g kg-1"

Table 6. Units for GLU, TP and THO must be "mg dL-1"

Lines 149-152. Specify the volume of serum sample used for these assays.

Author Response

Reviewer 1

The manuscript has been improved taking into account the reviewers' comments. Authors provide convincing explanations to each of the comments and make adequate modifications in the manuscript.

Just only a couple of minor changes for considering:

lines 131. Units must be "mg L-1"

Answer: Corrected.

lines 122 and 117. Unist must be "g kg-1"

Answer: Corrected.

Table 6. Units for GLU, TP and THO must be "mg dL-1"

Answer: Corrected.

Lines 149-152. Specify the volume of serum sample used for these assays.

Answer: Following the reviewer’s suggestion, we have inserted the volume of serum sample as follows: at least 0.5 mL.

Reviewer 3 Report

Authors properly replied to my comments.

Author Response

Thank you. 

Reviewer 4 Report

No further comments. 

Author Response

Thank you.